# Mechanism of Action of Atypical Antipsychotic Drugs in Mood Disorders

**DOI:** 10.3390/ijms21249532

**Published:** 2020-12-15

**Authors:** Daniil Grinchii, Eliyahu Dremencov

**Affiliations:** Institute of Molecular Physiology and Genetics, Center of Biosciences, Slovak Academy of Sciences, Dúbravská Cesta 9, 840 05 Bratislava, Slovakia; Daniil.Grinchii@savba.sk

**Keywords:** serotonin, norepinephrine, dopamine, histamine, adenosine, trace amines, atypical antipsychotics, mechanism of action, receptor pharmacology

## Abstract

Atypical antipsychotic drugs were introduced in the early 1990s. Unlike typical antipsychotics, which are effective only against positive symptoms of schizophrenia, atypical antipsychotics are effective against negative and cognitive symptoms as well. Furthermore, they are effective not only in psychotic but also in affective disorders, on their own or as adjuncts to antidepressant drugs. This review presents the neural mechanisms of currently existing atypical antipsychotics and putative antipsychotics currently being investigated in preclinical and clinical studies and how these relate to their effectiveness in mood disorders such as depression, anxiety, and post-traumatic stress disorder (PTSD). Typical antipsychotics act almost exclusively on the dopamine system. Atypical drugs, however, modulate serotonin (5-HT), norepinephrine, and/or histamine neurotransmission as well. This multimodal mechanism of action putatively underlies the beneficial effect of atypical antipsychotics in mood and anxiety disorders. Interestingly, novel experimental drugs having dual antipsychotic and antidepressant therapeutic potential, such as histamine, adenosine, and trace amine-associated receptors (TAAR) ligand, are also characterized by a multimodal stimulatory effect on central 5-HT, norepinephrine, and/or histamine transmission. The multimodal stimulatory effect on central monoamine neurotransmission may be thus primarily responsible for the combined antidepressant and antipsychotic therapeutic potential of certain central nervous system (CNS) drugs.

## 1. Introduction

The first antipsychotic drug, chlorpromazine, a member of the phenothiazine family, was discovered serendipitously in 1951. Phenothiazines were the first drugs shown to be effective in easing the positive symptoms of schizophrenia; they were, however, less effective against negative and cognitive symptoms and had multiple adverse side effects. Later, butyrophenones, another group of typical antipsychotic drugs, were discovered, with the best known being haloperidol. Haloperidol, introduced into clinical practice in 1958, remains the most frequently prescribed antipsychotic drug. Butyrophenones had better effectiveness against the positive symptoms of schizophrenia, with fewer side effects, but were less effective in treating the negative and cognitive symptoms [1].

The first atypical antipsychotic, clozapine, was introduced into clinics in 1990. Atypical antipsychotics revolutionized pharmacotherapy of schizophrenia: they were found to be effective not only against the positive but also against negative symptoms of schizophrenia. In addition, they have fewer extrapyramidal motor side effects than phenothiazines and butyrophenones [2,3]. Atypical antipsychotics, in comparison with typical ones, have demonstrated better efficacy in elderly patients with schizophrenia [4]. When atypical antipsychotics entered clinical practice, they were found to improve not only the core symptoms of schizophrenia but also the affective symptoms of depression that are frequently present in schizophrenic patients. Interestingly, the efficacy of atypical antipsychotics against schizophrenia-associated depression was higher in male than in female patients [5]. Later, the use of atypical antipsychotic drugs, as monotherapy or as adjuncts to antidepressant and mood-stabilizing medicines, became common in clinical practice also for mood and anxiety disorders, not necessarily associated with psychosis [6,7,8].

In general, atypical antipsychotic drugs as a monotherapy are effective in bipolar, not unipolar depression [9,10,11]. In particular, clinical effectiveness as monotherapy in bipolar disorder has been shown for aripiprazole [12,13], asenapine [14], olanzapine, quetiapine [15], and risperidone [16]. Quetiapine, aripiprazole, olanzapine, and risperidone have also been shown to be effective as monotherapies for generalized anxiety disorder [17].

In unipolar depression, atypical antipsychotic drugs are usually applied in combination with antidepressant drugs, such as selective serotonin reuptake inhibitors (SSRIs) or dual serotonin and norepinephrine reuptake inhibitors (SNRIs) [18,19,20]. In particular, creditable effectiveness as adjuncts to antidepressant drugs in treatment-resistant depression was demonstrated for aripiprazole [21], brexpiprazole [22], and risperidone [23]. A recent study by Cohen and colleagues demonstrated an anxiolytic effect of brexpiprazole combined with the SSRI escitalopram in an animal model of post-traumatic stress disorder (PTSD), measured using the elevated plus maze (EPM) test [24].

Interestingly, novel experimental treatment compounds, such as histamine [25,26,27], adenosine [28,29,30], and trace amine-associated receptor ligands [31,32,33], demonstrate putative antipsychotic, as well as antidepressant efficiency. In this sense, these putative future drugs are similar to contemporary atypical antipsychotic drugs. In this review, the neural mechanisms that allow atypical antipsychotic drugs, both those currently existing and those at different stages of preclinical or clinical investigation, to be effective in psychotic, as well as in affective disorders, are described.

## 2. Mechanism of Action of Atypical Antipsychotics in Mood Disorders

### 2.1. Serotonergic Mechanisms

Classical antipsychotics, such as haloperidol, act almost exclusively on D_2_ receptors (Figure 1A) [34]:

As such, their direct interaction with the central 5-HT system is limited. This is a possible explanation for the lack of efficacy of classical antipsychotics in mood disorders, unless psychotic symptoms are also present, as in psychotic depression or in a major manic episode with psychotic features. Unlike typical antipsychotics, atypical antipsychotics have affinities to 5-HT_2A/2C_, 5-HT_1A/1C_, α_1,_ and/or α_2_-adrenergic receptors, comparable or even higher than those to D_2_ receptors (Figure 1B) [34,35,36]. Targeting these receptors allows atypical antipsychotics to alter the excitability of 5-HT neurons. This may explain, at least in part, the efficacy of atypical antipsychotics in mood disorders.

The first direct examination of the effect of an atypical antipsychotic drug on the excitability of 5-HT neurons was performed by Stark and colleagues [37] and by Dremencov and colleagues [38] in 2007. Stark et al. reported that aripiprazole reduced the firing activity of 5-HT neurons in the dorsal raphe nucleus (DRN). This aripiprazole-induced inhibition of excitability of 5-HT neurons was reversed by a selective antagonist of 5-HT_1A_ receptors, WAY100635, suggesting that the serotonergic effect of aripiprazole is primarily mediated via 5-HT_1A_ receptors [37]. A similar acute inhibitory effect on 5-HT neurons was reported for risperidone; acute administration of paliperidone, however, did not alter 5-HT neuronal firing activity. In contrast to aripiprazole, risperidone-induced inhibition of 5-HT neurons was only partially reversed by WAY100635; complete reversal was observed after co-administration of WAY100635 and desipramine. A combined 5-HT_1A_ serotonergic/α_1_-adrenergic mechanism for the suppressive effect of risperidone on the excitability of 5-HT neurons was, therefore, suggested [38].

A sustained effect of atypical antipsychotic drugs may differ from the acute effect of the same medicines. For example, subchronic and chronic administration of aripiprazole stimulated the firing activity of 5-HT neurons. Co-administered with the SSRI escitalopram for two days, aripiprazole reverses escitalopram-induced inhibition of 5-HT neurons [39]. A similar stimulatory effect on 5-HT neurons was reported after two and fourteen days of brexpiprazole administration [40]. Sustained asenapine also boosted the excitability of 5-HT neurons, but after two and not fourteen days of treatment [41]. Sustained paliperidone did not alter the excitability of 5-HT neurons, while sustained risperidone inhibited it. Although paliperidone did not alter 5-HT neuronal firing activity on its own, it reversed two-day escitalopram-induced inhibition of 5-HT neurons. Risperidone, co-administered with escitalopram for two days, had an inhibitory effect on 5-HT neuronal firing activity similar to that of mono administration of each of these drugs (Figure 2) [38].

Risperidone and aripiprazole, despite their inhibitory effect on the firing activity of 5-HT neurons, increased cortical 5-HT levels each on its own and potentiated an escitalopram-induced increase in 5-HT concentrations in the rat frontal cortex, measured using in vivo microdialysis [42,43]. Furthermore, risperidone enhanced the anxiolytic and antidepressant-like behavioral effect of escitalopram, measured using the EPM and forced swim tests (FST), respectively. The potentiating impact of risperidone on the antidepressant-like behavioral effect of escitalopram was abolished by the administration of WAY100635 [44]. Thus, the antidepressant-like effect of risperidone, and perhaps of other atypical antipsychotics, is at least in part 5-HT-dependent.

The inhibition of 5-HT reuptake is the pharmacological mechanism of action of the majority of clinically used antidepressant drugs. This mechanism is shared by several tricyclic drugs, such as imipramine and clomipramine, as well as by SSRIs, SNRIs, and the last-generation triple 5-HT, norepinephrine, and dopamine reuptake inhibitors. The highly selective, last-generation SSRIs, such as citalopram and escitalopram, elevate extracellular 5-HT levels within minutes after their intake [45,46]. The onset of their clinical therapeutic effect, however, is not observed earlier than after two weeks of sustained drug administration. The inhibition of 5-HT neuronal firing activity is one of the mechanisms putatively responsible for the delayed response to antidepressant drugs. Thus, SSRIs, SNRIs, and triple 5-HT, norepinephrine and dopamine reuptake inhibitors first increase extracellular 5-HT levels in the raphe nuclei, where 5-HT transporters (SERTs) are densely expressed. This leads to the activation of somatodendritic 5-HT_1A_ autoreceptors and to the inhibition of the firing activity of 5-HT neurons [45,47,48,49,50,51,52,53]. As a result, the net 5-HT transmission, especially at the nerve terminal level, remains unchanged. Only after at least two weeks of treatment, when somatodendritic 5-HT_1A_ autoreceptors start to desensitize, and 5-HT neuronal firing activity recovers to basal levels, does net 5-HT transmission by SSRIs start to increase, and the therapeutic effect on behavior begins to be observed [54,55,56,57].

The therapeutic effect of some atypical antipsychotics, such as aripiprazole, brexpiprazole, and asenapine, as monotherapy in mood and anxiety disorders, might therefore be explained, at least in part, by the ability of these drugs to stimulate the firing activity of 5-HT neurons. The beneficial effect on the mood of quetiapine might be explained by the ability of this drug to increase the sensitivity of postsynaptic hippocampal 5-HT_1A_ receptors to 5-HT [58]. The beneficial effect of certain atypical antipsychotic drugs, such as aripiprazole and paliperidone, as adjuncts to SSRIs in treatment-resistant depression, may be explained by the ability of these drugs to reverse the SSRI-induced inhibition of 5-HT neuronal firing activity, especially at the early stages of SSRI treatment. The mechanisms allowing atypical antipsychotics to boost 5-HT excitability or to reverse SSRI-induced inhibition of 5-HT neurons are not yet completely understood. This mechanism perhaps involves rapid sensitization of 5-HT_1A_ autoreceptors and/or sensitization of α_1_-adrenergic receptors in the DRN [6,7].

### 2.2. Noradrenergic Mechanisms

Norepinephrine is a catecholamine that acts both as a hormone and as a central and peripheral neurotransmitter. In the CNS, norepinephrine is responsible for alertness, concentration, and energy [59]. Since fatigue and difficulty concentrating are frequent symptoms of depression [60], norepinephrine is likely to be involved in the pathophysiology of this disorder. Some antidepressant drugs, such as SSRIs, have a suppressing effect on norepinephrine transmission. For example, citalopram and escitalopram both inhibit the excitability of norepinephrine neurons of rat locus coeruleus (LC), escitalopram after two, and citalopram after fourteen days of treatment [38,61]. The more rapid effect of escitalopram, compared to citalopram, on the excitability of LC norepinephrine neurons is putatively explained by its higher selectivity and efficacy as a 5-HT reuptake inhibitor [45,46]. Unlike SSRI-induced inhibition of 5-HT neurons of the DRN, which disappears after sustained (two weeks or more) SSRI administration due to desensitization of somatodendritic 5-HT_1A_ autoreceptors, SSRI-induced inhibition of norepinephrine neurons of the LC remains in force even after prolonged treatment [38,39,61,62,63]. SSRI-induced inhibition of norepinephrine neuronal firing activity is apparently mediated, at least in part, via 5-HT_2A_ receptors [64,65].

It has been reported that the typical antipsychotic drug haloperidol does not alter the excitability of norepinephrine neurons of the LC on its own and does not alter the SSRI escitalopram-induced inhibition of firing activity of LC norepinephrine neurons [61]. The atypical antipsychotic drug olanzapine, however, stimulated the excitability of norepinephrine neurons of the LC after acute administration [66]. Increased activity of norepinephrine neurons was found after sub-chronic and chronic administration of aripiprazole [39], quetiapine [58], and brexpiprazole; brexpiprazole also increased sensitivity of hippocampal α_2_-adrenoceptors to norepinephrine [40]. The ability of aripiprazole combined with escitalopram to enhance cortical norepinephrine levels was also confirmed using in vivo microdialysis [43]. Asenapine boosted norepinephrine neuronal excitability after fourteen, but not after two days of treatment [41]. In contrast, acute, subchronic, and chronic administration of risperidone and paliperidone did not alter the norepinephrine neuronal firing activity in the LC [38,61,67].

In 2007, Dremencov and colleagues performed a pharmacological dissection of the effect of risperidone, alone or in combination with escitalopram, on the excitability of norepinephrine neurons of the LC [61]. Since risperidone is a potent 5-HT_2C_, D_2_, 5-HT_2A_, and α_2_-adrenoceptor antagonist, selective antagonists of these receptors (SB 242084, haloperidol, M100907, and idazoxan, respectively) were administered, alone or in combination with escitalopram, to assess whether their effect is similar to that of risperidone (Figure 3).

As already stated above, risperidone on its own did not alter norepinephrine firing activity, nor did SB 242084, haloperidol, or M100907. The only idazoxan increased the firing activity of 5-HT neurons. Escitalopram, also, as stated above, robustly suppressed the excitability of norepinephrine neurons of the LC. Neither SB 242084 nor haloperidol altered the escitalopram-induced inhibition of norepinephrine neurons. M100907 and idazoxan successfully reversed the inhibition of norepinephrine neurons induced by escitalopram. Risperidone not only reversed the escitalopram-induced excitability of norepinephrine neurons but also boosted the firing activity of norepinephrine neurons above the values recorded in control rats. Paliperidone [38], aripiprazole [39], and quetiapine [58] also reversed the SSRI-induced inhibition of norepinephrine neurons; these drugs, however, did not boost the firing of norepinephrine neurons to values higher than in control animals.

In summary, the beneficial effect of some atypical antipsychotics, such as aripiprazole, quetiapine, and brexpiprazole in mood disorders may be explained, at least partially, by their ability to enhance the excitability of norepinephrine neurons via an α_2_-adrenoceptor-mediated mechanism. The beneficial effect of antipsychotic drugs as adjuncts to SSRIs may be due, at least in part, to their ability to reverse the SSRI-induced inhibition of norepinephrine neuronal firing activity [6,68,69].

### 2.3. Dopaminergic Mechanisms

Dopamine is a brain catecholamine transmitter primarily responsible for reward and motivation [59]. Since anhedonia is a primary symptom of depression [60], dopamine is likely to be involved in the pathophysiology of this illness. At least some SSRIs have an inhibitory effect on dopamine transmission (VTA). The inhibitory effect of sustained SSRIs, such as escitalopram and citalopram, was demonstrated by Dremencov and colleagues; this effect was mediated, at least in part, via 5-HT_2C_ receptors [70]. Similar to the SSRI-induced inhibition of norepinephrine neurons and unlike SSRI-induced suppression of 5-HT neurons, the inhibition of dopamine neurons by SSRIs remains in force even after prolonged SSRI administration [39,70,71]. It is thus possible that the lack of adequate response to SSRIs, at least in some patients, can be mediated, at least in part, via the 5-HT_2C_ receptor-mediated inhibition of firing activity of 5-HT neurons.

It was reported that chronic administration of asenapine increased the density of spontaneously active dopamine neurons in the VTA while firing parameters remained unchanged. Asenapine, administered for two days, partially reversed the inhibition of dopamine neuronal firing activity induced by apomorphine, an agonist of D_2_ receptors. This effect was lost after chronic asenapine administration, suggesting adaptive changes leading to D_2_ receptor sensitization [41]. Aripiprazole, however, did not alter the excitability of dopamine neurons on its own. Nevertheless, when it was co-administered with escitalopram for two or fourteen days, aripiprazole reversed escitalopram-induced inhibition of excitability of dopamine neurons of the VTA [39]. It can thus be summarized that some atypical antipsychotics, such as asenapine, are beneficial in mood disorders because of their ability to stimulate mesolimbic dopamine transmission by increasing the density of spontaneously active dopamine neurons in the VTA. This beneficial mechanism putatively involves desensitization of D_2_ autoreceptors. Other atypical antipsychotic drugs, such as aripiprazole, are beneficial as adjuncts to SSRIs because of their ability to reverse SSRI-induced inhibition of dopamine neuronal firing activity via a mechanism putatively involving 5-HT_2C_ receptors activity [6,68,69].

### 2.4. Histaminergic Mechanisms

Some atypical antipsychotics, such as clozapine, olanzapine, and quetiapine, bind to histamine-1 (H_1_) receptors with an affinity comparable to that for D_2_, 5-HT_2A/2C_, and α_2_-adrenoceptors [34,35,36]. Interaction of CNS drugs with histamine receptors has traditionally been seen as being responsible for the side effects of these drugs on the immune system rather than for their therapeutic potential. However, since central histamine is involved in sleep, cognition, memory, and emotions [72,73], the therapeutic potential of histamine receptors has begun to attract attention [74].

In vivo interaction between histamine and other monoamines (5-HT, norepinephrine, and dopamine) was demonstrated in electrophysiological and microdialysis studies by Flik and colleagues. An increase in brain histamine levels, induced by a partial agonist of histamine-3/4 (H_3/4_) receptors, thioperamide [75], was found to lead to the subsequent elevation of extracellular 5-HT, norepinephrine, and dopamine concentrations in the prefrontal cortex (PFC) and hypothalamus [76].

Even though thioperamide increased brain dopamine, as well as 5-HT and norepinephrine concentrations, it stimulated the firing activity of dopamine, but not 5-HT and norepinephrine neurons. Immepip, an antagonist of H_3_ receptors, inhibited dopamine (but not 5-HT and norepinephrine) neurons on its own and reversed thioperamide-induced stimulation of dopamine neuronal firing activity. The stimulatory effect of thioperamide on the excitability of dopamine neurons was mimicked by a direct iontophoretic administration of histamine into the VTA. It was thus suggested that thioperamide increases extracellular histamine levels in certain brain areas, including the VTA, leading to the stimulation of dopamine-secreting neurons therein via a mechanism potentially involving histamine-1 (H_1_) receptors [76]. It is, therefore, possible that the antagonists of H_3_ and/or agonists of H_1_ receptors may be beneficial in mood and psychotic disorders, on their own and/or as an adjunct to antipsychotic and/or antidepressant drugs, because of their ability to stimulate brain monoamine transmission (Figure 1C).

It was indeed reported that thioperamide decreased rat immobility during the FST in a dose-dependent manner [77], indicating that H_3_ receptor blockers may exert an antidepressant-like effect. Another study reported a memory-facilitating effect of thioperamide [78]. In contrast, immepip decreased social isolation-induced vocalization in guinea pig pups and aggression in mice [79], suggesting an anxiolytic-like effect of H_3_ receptor agonists.

Even though the antidepressant-like effect of H_3_ receptor ligands is putatively mediated via the dopamine system [76,80], other signaling systems in the CNS may also play a role. For instance, intracerebroventricular histamine induces adrenocorticotropic hormone (ACTH) and prolactin release from the pituitary [81]. In contrast, H_3_ receptor agonists inhibited ACTH and prolactin response to restraint stress and the ACTH response to an immune challenge by lipopolysaccharide (LPS) in rats [82]. Histamine-induced prolactin release was attenuated by blockers of the 5-HT_2_ receptor [81]. It is thus likely that histamine and its interaction with 5-HT are important in the regulation of the release of some stress hormones.

### 2.5. Purinergic Mechanisms

Hence, far, receptors to purines, such as adenosine-2A (A_2A_) receptors, are not targeted by the contemporary antipsychotic drugs, either typical or atypical. However, there is evidence for the involvement of these receptors in both psychotic and affective disorders and for their potential role as a target for next-generation CNS medicines. An antipsychotic-like effect of an agonist of A_2A_ receptors, CGS 21680, in laboratory primates was demonstrated by Andersen and colleagues in 2002 [83]. This antipsychotic-like effect of CGS 21,680 was explained by its ability to attenuate the affinity of postsynaptic D_2_ receptors in the striatum to dopamine [84,85,86]. It has recently been reported that A_2A_ and D_2_ receptors interact on the molecular level: these two receptors are capable of forming heterodimers. Furthermore, A_2A_-D_2_ heterodimers have a different signal transduction mechanism (Gα_Q/Z_ protein-mediated) than A_2A_ (Gα_S_ protein-mediated) or D_2_ (Gα_I/O_ protein-mediated) receptors [87].

In vivo microdialysis and electrophysiological examination of the effect of an agonist (CGS 216,820) and an antagonist (ZM 241,385) of A_2A_ receptors, alone or in combination with the classic antipsychotic drug haloperidol, on the excitability of dopamine and norepinephrine neurons and on concentrations of catecholamines in the PFC and nucleus accumbens (NAcc) was performed by Dremencov and colleagues in 2017 [88]. It was found that an antagonist of A_2A_ receptors, CGS 216820, inhibited the firing activity of norepinephrine neurons of the LC and dopamine neurons of the VTA, and an antagonist of A_2A_ receptors, ZM 241385, reversed CGS 216,820-induced inhibition of norepinephrine and dopamine neurons. ZM 241385 did not alter extracellular levels of catecholamines on its own, but it did potentiate the effect of haloperidol on NAcc dopamine and PFC norepinephrine. It is thus possible that A_2A_ receptor ligands may be beneficial in affective and psychotic disorders, on their own and/or as an adjunct to antipsychotic and/or antidepressant drugs, because of their ability to modulate central catecholamine transmission (Figure 1C).

Behavioral studies in rodents support the hypothesis that A_2A_ receptor ligands might have antipsychotic [83], as well as antidepressant-like effects. Yacoubi and colleagues [89] reported that the A_2A_ receptor blockers SCH 58261 and KW 6002 had anxiolytic and antidepressant-like effects, measured by the tail suspension test. On the other hand, Kaster and co-authors [90] reported an antidepressant-like effect of adenosine, measured by the FST. This antidepressant-like effect of adenosine was diminished by pretreatment with nonselective (e.g., caffeine), as well as selective (ZM 241,385) blockers of A_2A_ receptors. A later study [91] showed that the antidepressant-like effect of adenosine was blocked by WAY100635, suggesting a critical role of adenosine-5-HT interactions in the putative antidepressant-like effect of the A_2A_ receptor ligands. However, the antidepressant-like effect of adenosine was also blocked by pretreatment with nonselective δ/µ- (naloxone) and selective δ- (naltrindole) and µ- (clocinnamox) opioid-receptor antagonists [90], indicating an important role for the endogenous opioid system in the beneficial effect on the mood of adenosine.

### 2.6. Trace Aminergic Mechanisms

Trace amines are biological molecules that are present in the mammal brain in trace concentrations. The trace amines include phenethylamines (phenethylamine, n-methylphene-thylamine, phenylethanolamine, m-and p-tyramine, 3-methoxytyramine, n-methyltyramine, m- and p-octopamine, and synephrine), thyronamines (3-iodothyronamine), and tryptamines (tryptamine). Trace amines are closely related to the “classical” monoamines from the structural and metabolic points of view [92].

Because of their negligible concentrations, trace amines were considered not to have any important biological function. However, the discovery of trace amino acid receptors (TAARs) in 2001 indicated that trace amines are important signaling molecules, acting as brain neurotransmitters [93]. Five TAARs have so far been identified: TAAR1, TAAR2, TAAR5, TAAR6, and TAAR8. All TAARs are GPCRs. TAAR1 is believed to be G_S_, TAAR5-G_S_ and/or G_Q_, and TAAR8-G_I_-coupled [94].

TAAR1 receptors are of special interest as a target for future antidepressant and antipsychotic drugs (Figure 1C). Three lines of evidence support this hypothesis. First, TAAR1 receptors are densely expressed in the brain, and particularly in the DRN and VTA [95]. Secondly, some TAAR1 ligands have shown antidepressant- and antipsychotic-like effects in rodents and in primates. Thus, the full agonist RO5256390 decreases the immobility time of rats during the FST, and both RO5256390 and the partial TAAR1 agonist RO5263397 improves the differential reinforcement of monkeys for low-rate (DRL) scores [31]. Third, TAAR1 ligands have been shown to modulate monoamine neurotransmission. RO5256390 [31] and another agonist of TAAR1, RO5166017 [96], were shown to inhibit ex vivo excitability of 5-HT and dopamine neurons in brain slices. Consistently, knockout mice lacking TAAR1 showed decreased ex vivo excitability of 5-HT and dopamine neurons. It was also shown that the excitatory effect of TAAR1 agonists on dopamine neurons is mediated, at least in part, via the suppression of their γ-aminobutyric acid (GABA)-mediated inhibition [95]. The same study showed, using in vivo microdialysis, that TAAR1-knockout mice have increased concentrations of catecholamines in the NAcc and 5-HT levels in the PFC. More recently, the inhibitory effect of acutely administered RO5256390 on 5-HT and dopamine neuronal firing activity was observed in in vivo conditions; RO5166017 reversed RO5256390-induced suppression of 5-HT and dopamine neuronal firing activity. [97]. The antidepressant-like behavioral effects of TAAR1 ligands in rodents, reported in previous studies [31], may thus be explained by the ability of these compounds to modulate the excitability of 5-HT and dopamine neurons.

## 3. Summary

The experimental findings on the effect of atypical antipsychotic drugs, administered alone or in combination with SSRIs, are summarized in Table 1. The stimulatory effect of some atypical drugs, such as asenapine, aripiprazole, and brexpiprazole, on 5-HT neuronal firing activity and their diminishing effect on subchronic-SSRI-induced inhibition of 5-HT neurons is putatively mediated via the blockade of 5-HT_1A_ autoreceptors and induction of their rapid desensitization. The diminishing effect of paliperidone subchronic-SSRI-induced inhibition of 5-HT neurons may be mediated via 5-HT_1A/1C_ and/or α_1_-adrenoceptor-mediated mechanisms. The diminishing effects of atypical antipsychotic drugs on SSRI-induced inhibition of norepinephrine and dopamine neurons are 5-HT_2A_- and 5-HT_2C_-mediated, respectively. The α_2_-adrenoceptor antagonistic property of some atypical drugs, such as olanzapine and risperidone, contribute to their norepinephrine-mediated mechanism of action as well.

## 4. Conclusions

Atypical but not typical antipsychotic drugs enhance 5-HT, norepinephrine and/or dopamine transmission on their own and/or diminish the inhibition of firing activity of monoamine-secreting neurons induced by some antidepressant drugs. The evidence correlates well with clinical observations, suggesting that atypical but not typical antipsychotic drugs are effective against negative and cognitive symptoms of schizophrenia, as well as against affective symptoms in mood and anxiety disorders. It is thus possible that similar pathophysiological mechanisms underline the negative symptoms of schizophrenia and mood disorders and that effective treatment of both requires simultaneous boosting of 5-HT, norepinephrine, and dopamine transmission.

## Figures and Tables

**Figure 1 ijms-21-09532-f001:**
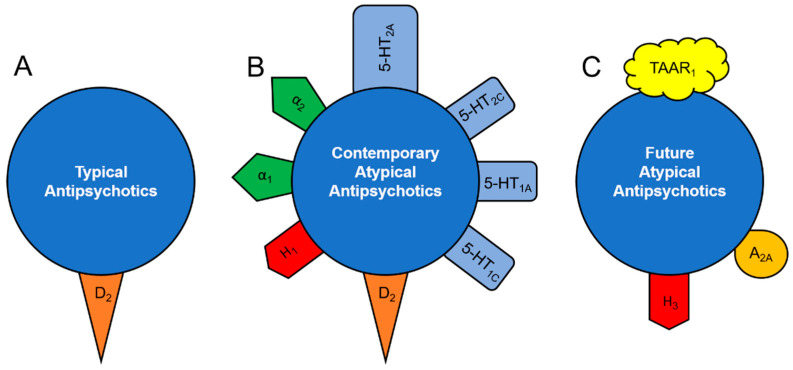
Receptor-binding profile of antipsychotic drugs. Typical antipsychotics (**A**) act almost exclusively as blockers of dopamine-2 (D_2_) receptors. All contemporary atypical antipsychotic drugs (**B**) are characterized by serotonin-2A (5-HT_2A_) antagonistic property, comparable to or even higher than their D_2_ blocking potential. Some atypical antipsychotics are also potent serotonin-1A (5-HT_1A_; aripiprazole), serotonin-1C (5-HT_1C_; clozapine, olanzapine, risperidone), histamine-1 (H_1_; olanzapine, quetiapine) and α_1_-(aripiprazole, clozapine, olanzapine, paliperidone, quetiapine) and α_2_-adrenergic (clozapine, olanzapine, paliperidone, quetiapine, risperidone) receptor blockers. Future drugs with dual antipsychotic and antidepressant therapeutic potential (**C**) may also target trace amine-associated receptor-1 (TAAR1), as well as histamine-3 (H_1_) and adenosine-2A (A_2A_) receptors.

**Figure 2 ijms-21-09532-f002:**
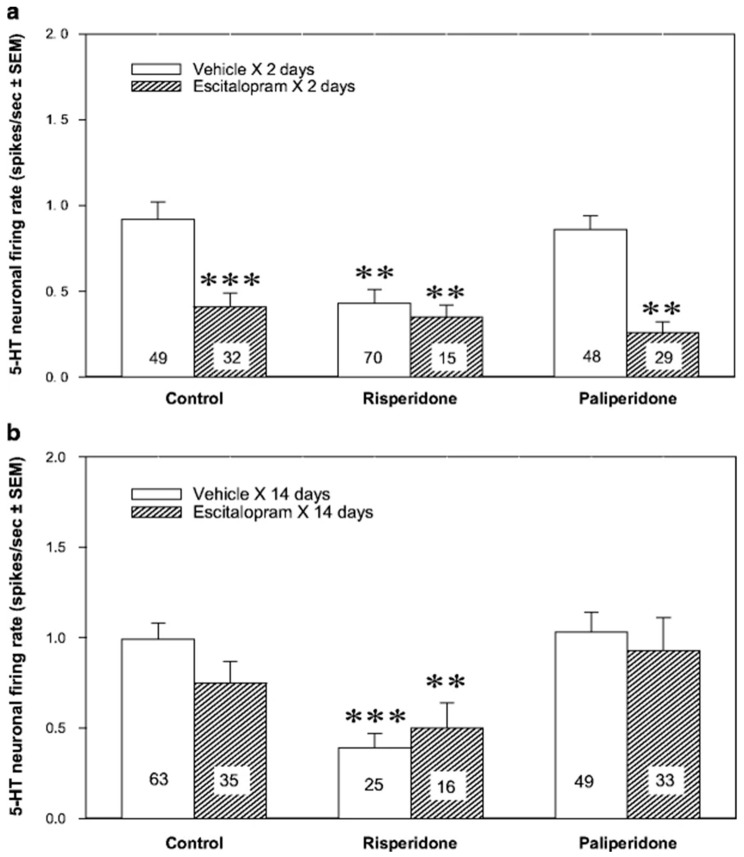
Serotonergic augmentation of the selective serotonin reuptake inhibitors (SSRI) escitalopram response by the atypical antipsychotic drug paliperidone. Effect of the subchronic (**a**) and chronic (**b**) risperidone or paliperidone, alone or in combination with escitalopram, on the excitability of 5-HT neurons of the dorsal raphe nucleus (DRN). The animals were implanted with minipumps containing the vehicle or escitalopram (10 mg/kg/day) for two or fourteen days and received the subcutaneous vehicle, risperidone or paliperidone injections (1 mg/kg each). After two days, there was a significant effect of escitalopram (F_df1243_ = 26.16, *p* < 0.001), risperidone and paliperidone (F_df2243_ = 3.99, *p* < 0.05), and escitalopram × risperidone/paliperidone interaction (F_df2243_ = 3.93, *p* < 0.05). After fourteen days, there was a significant effect of risperidone and paliperidone (_Fdf2221_ = 6.61, *p* < 0.01). The number of neurons recorded in each group is provided within the histograms. ** *p* < 0.01 and *** *p* < 0.001 in comparison with control animals. From Dremencov et al. 2007 [38]. Reused by permission of Springer Nature.

**Figure 3 ijms-21-09532-f003:**
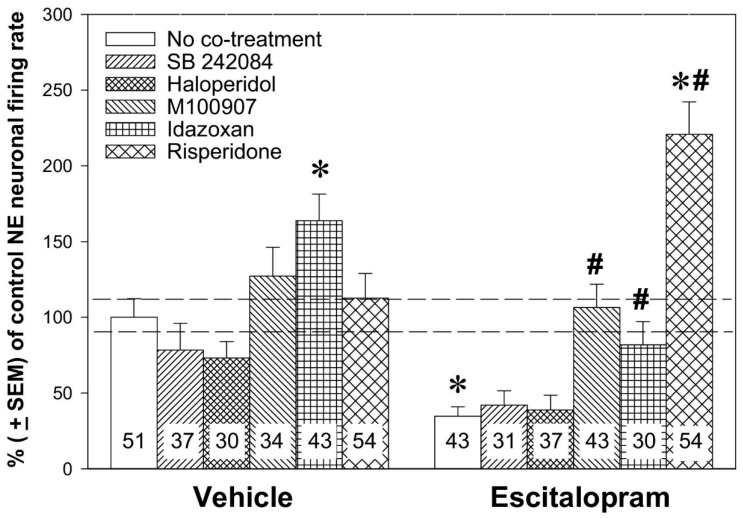
Noradrenergic augmentation of the SSRI escitalopram response by the atypical antipsychotic drug risperidone. The animals were implanted with minipumps containing vehicle or escitalopram (10 mg/kg/day; the main treatment) for two days and were subcutaneously (s.c.) co-treated with vehicle, risperidone (1 mg/kg/day), SB 242084 (0.5 mg/kg/day), haloperidol (0.1 mg/kg/day), M100907 (0.5 mg/kg/day), or idazoxan (1 mg/kg/day; co-treatments) for two days. Two-way ANOVA showed a significant effect of the main treatment (F_df1486_ = 5.41, *p* < 0.05), of the co-treatment (F_df5486_ = 17.18, *p* < 0.001), and treatment × co-treatment interaction (F_df5486_ = 11.07, *p* < 0.001). * *p* < 0.01, in comparison with control animals and # *p* < 0.05, in comparison with animals administered escitalopram alone, Bonferroni post hoc test. The number of neurons recorded in each group is provided within the histograms. From Dremencov et al. 2007 [61]. Reused by permission of Elsevier.

**Table 1 ijms-21-09532-t001:** Effects on contemporary and putative future atypical antipsychotic drugs, alone or in combination with antidepressants, on the excitability of 5-HT, norepinephrine (NE), and dopamine (DA) neurons.

Drug	Neurophysiological Effect	
	On its Own (Monotherapy)	In Combination with SSRIs
**Currently Used Atypical Antipsychotic Drugs**		
Aripiprazole	5-HT neurons ↓ (acute) 5-HT neurons ↑ (subchronic, chronic) NE neurons ↑ (subchronic, chronic)	SSRI-induced 5-HT inhibition (subchronic) ↓ SSRI-induced NE inhibition (subchronic, chronic) ↓ SSRI-induced DA inhibition (subchronic, chronic) ↓
Asenapine	5-HT neurons ↑ (subchronic) NE neurons ↑ (chronic) DA neurons ↑ (chronic)	
Brexpiprazole	5-HT neurons ↑ (subchronic, chronic) NE neurons↑ (subchronic, chronic)	
Olanzapine	NE neurons ↑ (acute)	
Paliperidone		SSRI-induced 5-HT inhibition (subchronic) ↓ SSRI-induced NE inhibition (subchronic, chronic) ↓
Risperidone	5-HT neurons↓ (acute, subchronic, chronic)	SSRI-induced NE inhibition (subchronic, chronic) ↓
Quetiapine	NE neurons ↑ (subchronic, chronic)	SSRI-induced NE inhibition (subchronic, chronic) ↓
**Putative Novel Atypical Antipsychotic Drugs**		
Thioperamide (H_3/4_ partial agonist)	DA neurons ↑ (acute)	
ZM 241385 (A2A antagonist)	DA neurons ↑ (acute)	
RO5263397 (TAAR1 partial agonist)	5-HT neurons ↑ (acute) DA neurons ↑ (acute)	

↑: stimulatory/increasing effect; ↓: inhibitory/diminishing effect.

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
