# Peer review of "Mechanism of Action of Atypical Antipsychotic Drugs in Mood Disorders"

_ijms, 2020, doi:10.3390/ijms21249532_

Round 1
Reviewer 1 Report
In the review article titled: “Mechanism of Action of Atypical Antipsychotic Drugs in Mood Disorders” co-authored by D. Grinchii and E. Dremencov, they comprehensively review and discuss the literature pertaining to the effectiveness and underlying pharmacological effects on neurotransmitter function in mood disorders. There is a scientific basis for this review, the different neurotransmitter mechanisms are discussed in a succinct manner, however, the novelty and importance of the review itself need to be made more evident and the review needs to be edited for English grammar and syntax by a Native or proficient speaker because as is, it is very difficult to get the message across with the grammatical and punctuation errors present (especially the abstract and intro).
In addition, I think the review can be significantly improved by including behavioral pharmacological studies, where relevant; ie. related effects on behavior in rodents (i.e. which neurotransmitter are implicated in and govern symptoms- depression, anxiety, cognition, etc.), in addition to discussion on neurotransmission. There are a couple of examples included in the discussion, but only for trace amines, p. 7 lines 290-293.
As the purinergic system and trace amines are discussed as a putative targets for mood disorders, exclusive of the action of typical and atypical antipsychotics, it makes sense to discuss other systems like glutamate, GABA or endocannabinoids.
Minor comments:
Keywords: these words are used to index the article on pubmed, etc. so I would recommend adding words like: atypical antispychotics, mechanism of action, receptor pharmacology
Abstract: The abstract should be more succinct and mention the novelty and the timeliness of the review- why should a scientist read it/ what new does it have to offer over previous reviews on the subject? For example, This review aims to discuss the respective neural mechanisms of currently existing atypical antipsychotics and putative antipsychotics currently being investigated in preclinical and clinical studies and how these relate to their effectiveness in mood disorders (namely…). Explicitly state the scope of the review. I would suggest removing the lines: “Blocking of 5-HT1A/1B autoreceptors, inducing their early desensitization, 17 and/or activation of α1-adrenoceptors, allow some atypical drugs to enhance 5-HT transmission. 18 Blocking of 5-HT2A/2C and/or α2-adrenoceptors enable some atypical antipsychotics to stimulate catecholamine transmission and/or diminish the inhibition of catecholamine neurons induced by some antidepressants. It is possible, that the activation of H1 and/or blocking of H3 boost monoamine transmission as well, via a mechanism involving stimulation of firing activity of dopamine neurons” and adding them to the body- this info is too detailed for the abstract. It is better to focus on summarizing the topics discussed (i.e. receptor pharmacology), the novelty and the scope.
Introduction:
Lines: 32-33, p.1: “they were effectiveness against negative and cognitive symptoms and had multiple aversive side effects” should be corrected to “they were less effective against negative and cognitive symptoms and had multiple adverse side effects”. Same for butyrophenones, “less effective” not “still effectiveness”.
- Use “effective” not “affective” to describe efficacy (e.g. line 41, p.1)
Subsections
- Line 73, p.2: “comparable” not “compatible”
- Line 74, p.2: Regarding the statement “This may explain the efficiency of atypical antipsychotics in mood disorders per se”. First, I think you are referring to their efficacy (not efficiency) and secondly, I would omit “per se” as you cannot exclude a contribution of other receptors in their efficacy (e.g. histamine, mentioned in the text). First generation antipsychotics are also effective antimanic agents. Alternatively, focus the statement on depressive symptoms and cite a reference.
- Line 84, p. 2: please correct: “c-administration”
- I believe Figure 1 is incorrect. It refers only to paliperidone effects on NE neuronal firing and not risperidone and paliperidone coadministered with escitalopram (Fig 5 in the cited reference).
- The first line of the summary, p. 8, line 305 mentions that typical antipsychotics are included in table 1 but this is not the case.
- Table 1’s title/legend is missing and there is a misplaced figure 2 legend underneath.
Reviewer 2 Report
This article reports the mechanism of action of atypical antipsychotics in mood disorders.
Overall it is a good paper, but I wish I could underline some remarkable points:
- The English language has to be checked in some sentences
- It would be more appreciated to the reader if the authors added a figure reporting the differences among the atypical antipsychotics on the various receptors
- Are there any gender differences and age-related changes in the response to atypical antipsychotics in mood disorders?
Round 2
Reviewer 1 Report
The authors have addressed all my comments and made appropriate changes to the manuscript. The only comment I have is that I would recommend adding a concluding statement to the abstract.